# Fatty Pancreas Is a Risk Factor for Pancreatic Cancer: A Systematic Review and Meta-Analysis of 2956 Patients

**DOI:** 10.3390/cancers15194876

**Published:** 2023-10-07

**Authors:** Mónika Lipp, Dorottya Tarján, Jimin Lee, Ádám Zolcsák, Eszter Szalai, Brigitta Teutsch, Nándor Faluhelyi, Bálint Erőss, Péter Hegyi, Alexandra Mikó

**Affiliations:** 1Institute of Pancreatic Diseases, Semmelweis University, 1083 Budapest, Hungary; lipp.monika@semmelweis.hu (M.L.); tarjan.dorottya@semmelweis.hu (D.T.); eross.balint@semmelweis.hu (B.E.); hegyi.peter@semmelweis.hu (P.H.); 2Centre for Translational Medicine, Semmelweis University, 1085 Budapest, Hungary; ji.lee@stud.semmelweis.hu (J.L.); zolcsak.adam@med.semmelweis-univ.hu (Á.Z.); szalai.eszter@semmelweis.hu (E.S.); teutsch.brigitta@semmelweis.hu (B.T.); faluhelyi.nandor@pte.hu (N.F.); 3Institute for Translational Medicine, Medical School, University of Pécs, 7624 Pécs, Hungary; 4Medical School, Semmelweis University, 1085 Budapest, Hungary; 5Department of Biophysics and Radiation Biology, Semmelweis University, 1094 Budapest, Hungary; 6Department of Restorative Dentistry and Endodontics, Semmelweis University, 1088 Budapest, Hungary; 7Department of Medical Imaging, Medical School, University of Pécs, 7624 Pécs, Hungary; 8Translational Pancreatology Research Group, Interdisciplinary Centre of Excellence for Research Development and Innovation, University of Szeged, 6725 Szeged, Hungary; 9Department of Medical Genetics, Medical School, University of Pécs, 7624 Pécs, Hungary

**Keywords:** pancreatic steatosis, pancreatic adenocarcinoma, risk factors, metabolic syndrome, obesity

## Abstract

**Simple Summary:**

Pancreatic cancer (PC) is infamous for its silent and lethal nature, making it one of the leading causes of cancer-related deaths worldwide. The early detection of PC is critical to achieve a favorable prognosis, as it typically goes unnoticed until advanced stages. Fatty pancreas (FP) is often associated with PC, but the causal relationship has not been clearly defined. With this systematic review and meta-analysis, we aim to strengthen the connection and emphasize the potential risks of FP to the development of PC. Based on our analysis, our findings exhibit a clear risk of PC in the presence of FP, advocating for future prospective studies to be performed in order to solidify the relationship.

**Abstract:**

Pancreatic cancer (PC) is one of the most lethal cancers worldwide. Recently, fatty pancreas (FP) has been studied thoroughly, and although its relationship to PC is not fully understood, FP is suspected to contribute to the development of PC. We aimed to assess the association between PC and FP by conducting a systematic review and meta-analysis. We systematically searched three databases, MEDLINE, Embase, and CENTRAL, on 21 October 2022. Case–control and cross-sectional studies reporting on patients where the intra-pancreatic fat deposition was determined by modern radiology or histology were included. As main outcome parameters, FP in patients with and without PC and PC in patients with and without FP were measured. Proportion and odds ratio (OR) with a 95% confidence interval (CI) were used for effect size measure. PC among patients with FP was 32% (OR 1.32; 95% CI 0.42–4.16). However, the probability of having FP among patients with PC was more than six times higher (OR 6.13; 95% CI 2.61–14.42) than in patients without PC, whereas the proportion of FP among patients with PC was 0.62 (95% CI 0.42–0.79). Patients identified with FP are at risk of developing PC. Proper screening and follow-up of patients with FP may be recommended.

## 1. Introduction

Fatty pancreas (FP) is referred to as the Pandora’s box of pancreatology [1], as this widespread condition may be an overlooked sign of pancreatic cancer (PC). Pancreatic cancer is known as the most common asymptomatic as well as most lethal cancer in developed countries [2,3]. It is predicted that PC will be the leading cause of all cancer-related deaths by 2050 [4]. The connection between FP and PC has not been extensively studied; however, it would open a new horizon for individuals at high risk for PC.

The mortality rate of PC can be attributed to its late presentation and resistance to chemotherapy [5]. Hur et al. [6] discovered that timely detection and tumor size are the strongest predictors of survival, supporting the importance of early PC detection.

Several risk factors have been identified to play a role in the development of PC. Among these, genetic mutations are responsible for 10–15% of all PC cases, for which there are screening protocols for high-risk individuals [5,7]. Additionally, high alcohol intake, diabetes mellitus, obesity, and chronic pancreatitis (CP) are among the other risk factors that significantly contribute to the development of PC [8]. Obesity has a major impact on the progression of pancreatic diseases [9]. Furthermore, a meta-analysis has found that CP patients have an eightfold increased risk of developing PC five years after the initial diagnosis [10].

At the time of diagnosis, only 10–20% of patients with PC were found in resectable or localized stages [11]. It is crucial to develop a precise screening protocol for the high-risk population unaffected by genetic mutations to facilitate early PC detection.

Due to the high mortality rate as a result of the late diagnosis of PC, non-genetic factors should be determined for early PC detection. FP is a possible option as it can be detected earlier than icterus, which is one of the most common initial features of PC. Fatty pancreas is defined as a significant fatty infiltration of the pancreatic parenchyma and has been associated with increasing age, obesity, metabolic syndrome, non-alcoholic fatty liver disease, and PC [12,13,14,15]. The prevalence of FP varies across the studies from 11% to 35% [16,17]. Obesity stands out as a significant risk factor for PC [18]. Furthermore, increased pancreatic adiposity contributes to the initiation of PC from FP [14]. Studies by Hori et al. [19] conducted on animals have revealed that obesity and hyperlipidemia exacerbate fatty infiltrated pancreas, consequently accelerating the progression of PC. This process is associated with the increased expression of adipocytokines and genes related to cell proliferation within the pancreatic tissue. Hence, the local production of adipocytokines from adipose tissues in an environment rich in adipose tissue is closely linked to the development of PC [20].

The lack of guidelines for reporting intra-pancreatic fat deposition and the need for early predictors of PC necessitated our analysis. We hypothesized that FP contributes to the development of PC. We aimed to determine the importance of FP as an early risk factor for the development of PC.

## 2. Materials and Methods

### 2.1. Protocol

The Preferred Reporting Items for Systematic Reviews and Meta-Analyses (PRISMA) 2020 guideline [21] (see Appendix A) and Cochrane Handbook [22] were used. PROSPERO registration no. is CRD42022369017.

### 2.2. Eligibility Criteria

Studies were included in which the pancreas of an adult human population was assessed using modern radiology (endoscopic ultrasound (EUS), magnetic resonance imaging (MRI), and computed tomography (CT)) or histology to determine the presence of intra-pancreatic fat deposition in PC and/or control patients. Population, Exposure, Comparator, and Outcome(s) (PECO) and Condition, Context, and Population (CoCoPop) frameworks were used to identify the articles eligible for this study. We approached the research question from two directions. In analysis A, FP was defined by modern radiology or histology (P). Articles were eligible if they reported whether patients had PC (E) or not (C) and assessed the outcome of the ratio of PC (O). In analysis B, patients with PC were examined (P) for the presence (E) or absence of FP (C), and the ratio of FP (O) was determined. Moreover, we ascertained the prevalence of FP (Co), where FP was defined by modern radiology or histology (Co) in patients with PC (Pop) in analysis B. All case reports, conference abstracts, reviews, narrative summaries, systematic reviews, and meta-analyses were excluded. No language filter was used.

### 2.3. Search Strategy

The search key for this review can be found in Document S1. The systematic search was performed in three electronic databases: MEDLINE (via PubMed), Embase, and Cochrane Central Register of Controlled Trials (CENTRAL) without any filters on 21 October 2022. After full-text selection, citation chasing was performed [23] to identify any additional research that may not have been found during the systematic search.

### 2.4. Selection Process

Articles collected from the systematic search were processed using the Rayyan web software [24] and EndNote 20 software (Clarivate Analytics, Philadelphia, PA, USA). After duplicate removal, two independent review authors (ML and DT) performed the selection according to the criteria. The senior review author (AM) resolved any disagreements between the two reviewers. Cohen’s kappa was calculated to quantify the degree of agreement between the two investigators.

### 2.5. Data Collection

From the eligible articles, two independent investigators (ML and DT) extracted the following data into a purpose-designed Excel sheet (Office 365, Microsoft, Redmond, WA, USA): overall demographic data (age, gender, and number of individuals); characteristics of individuals with PC and without PC (number of individuals, age, gender, FP, diabetes, chronic pancreatitis, fibrosis, hyperlipidemia, hypertension, family history of PC, and body mass index (BMI)). When available, odds ratio (OR) and their 95% confidence interval (CI) were recorded. The senior review author (AM) resolved all disagreements between the two reviewers. When necessary, the authors of the respective studies were contacted via email to request additional information to support our study.

### 2.6. Study Risk of Bias Assessment

Two authors (ML and DT) independently assessed the risk of bias using tools developed by the Joanna Briggs Institute (JBI) and the Quality of Prognostic Studies tool (QUIPS) [25,26]. The results of the assessment were graphically represented. Disagreements were resolved by consensus and the involvement of the senior review author (AM).

### 2.7. Statistical Analysis

The detailed statistical analysis can be found in Document S2. As we assumed considerable between-study heterogeneity in all cases, we used a random effects model to pool effect sizes. Proportions and ORs with 95% CI were used for effect size measures. To determine the prevalence or OR, the total number of patients and those with the event of interest (in each group separately for OR) were extracted from every study. We reported the results as the odds of an event of interest in the experimental group versus the odds of an event of interest in the control group. OR results were considered statistically significant if the pooled CI did not contain the null value (OR = 1). We summarized the findings of the meta-analysis on forest plots. Where applicable—when the study number was large enough and not too heterogeneous—we also reported the prediction intervals (i.e., the expected range of effects of future studies) of results. Additionally, between-study heterogeneity was described by Higgins and Thompson’s I^2^ statistics [27]. Small study publication bias was assessed by visual inspection of Funnel plots and calculating the p-value of the Peters test [28] for proportion effect size and the Harbord test p-value [29] for OR effect size. We planned to assume possible small study bias if the p-value was less than 10%. However, we kept in mind that the test had limited diagnostic assessment below 10 studies. Potential outlier publications were explored using different influence measures and plots following the recommendation of Harrer et al. [30].

All statistical analyses were performed with R v4.2.2 [31] using the meta [32] package v6.1.0 for basic meta-analysis calculations and plots and the dmetar [33] package v0.0.9000 for additional influential analysis calculations and plots.

### 2.8. Quality of Evidence

Grading of Recommendations Assessment, Development, and Evaluation (GRADE) workgroup recommendation [34] and the GRADE profiler (GRADEpro) tool [35] were applied.

## 3. Results

### 3.1. Study Selection of the Included Studies

A total of 7077 studies were identified during the initial search. Studies were selected according to eligibility criteria, resulting in 17 eligible articles with 2956 patients [36,37,38,39,40,41,42,43,44,45,46,47,48,49,50,51,52], although one was excluded due to reporting on overlapping populations [14]. No further articles were identified after citation chasing. The PRISMA flowchart (Figure 1) illustrates the literature screening process of the studies using the calculated Cohen’s kappas.

### 3.2. Study Characteristics

The main characteristics of the included studies are listed in Table 1 and Table 2. The publication year ranged from 2011 to 2022. Our analysis included 16 retrospective studies [34,35,36,37,38,39,40,41,42,43,44,45,46,47,48,49] and 1 prospective study [33]. Most studies investigated the Japanese population [34,38,39,40,41,43,44,45,47,48]. Three studies examined the presence of PC among patients with FP [33,34,35], nine studies evaluated the presence of FP among patients with PC [36,37,38,39,40,41,42,43,44], and we found fourteen studies where we examined the proportion of FP among PC patients [36,37,38,39,40,41,42,43,44,45,46,47,48,49]. EUS was used in six studies [33,35,36,40,42,44], CT in five studies [37,43,45,47,49], and histology in three studies [38,39,46] for the diagnosis of FP. In a few studies, the diagnosis was made with histology combined with imaging modalities: EUS [48], MRI [41], and CT [34]. We did not analyze secondary outcomes due to a lack of data. All studies were full-text articles and written in English [33,34,35,36,37,38,39,40,41,42,43,44,45,46,47,48,49].

### 3.3. No Relationship between Patients with FP and the Existence of PC

We found three studies [33,34,35] that investigated PC in the FP population (Figure 2). Although an increase of 32% in the OR would have clinical relevance, no significant relationship was found between patients with FP and the presence of PC (OR 1.32; 95% CI 0.42–4.16), I^2^ = 0.28 (95% CI: 0–0.93).

### 3.4. Fatty Pancreas Is Six Times More Common among Individuals Diagnosed with PC

We investigated the presence of FP among patients with PC (Figure 3). Data for the analysis were pooled from nine articles [36,37,38,39,40,41,42,43,44]. We found a significant association between patients with PC and the presence of FP (OR 6.13; 95% CI 2.61–14.42), I^2^ = 0.67 (95% CI: 0.34–0.84).

### 3.5. Fatty Pancreas Is Found in 62% of the PC Patients

The prevalence of FP among patients with PC was also investigated, and data were collected from 14 articles [36,37,38,39,40,41,42,43,44,45,46,47,48,49] (Figure 4). In our proportion analysis, FP was found in 62% of patients with PC (0.62; 95% CI 0.42–0.79), I^2^ = 0.89 (95% CI: 0.84–0.93).

### 3.6. Risk of Bias Assessment

The risk of bias assessment for each outcome is presented in Appendix A. In the studies in analysis A [33,34,35], as well as in those examining patients with PC and the presence of FP [36,37,38,39,40,41,42,43,44], the Quality of Prognostic Studies (QUIPS) tool was applied. The study attrition domain did not apply to most of the studies. Where the proportion of FP among PC patients was examined, the Joanna Briggs Institute (JBI) tool was applied. We used a 10% precision in the sample size calculation, resulting in three articles with adequate sample sizes. In domain nine, not all articles were applicable.

### 3.7. Publication Bias and Heterogeneity

Small study bias was not assessed for analysis A as there were only three studies. In the assessment of publication bias for studies of patients with PC where the presence of FP was examined (Appendix A), a small study bias may be present based on visual inspection of the funnel plot. On the other hand, one large study shows a similar or larger effect than small studies, and the Harbord test was not significant at a 10% level. Additionally, the results of the other four large studies also indicate significant results. For the proportion of FP in patients with PC (Appendix A), we can accept the hypothesis of having no small study bias based on visual inspection and the Peters test.

### 3.8. Certainty of Evidence

The certainty assessment of the three investigated outcomes shows very low to moderate certainty of the evidence, as presented in Appendix A. The reasons for downgrading the quality of evidence were due to serious inconsistencies and, in some cases, serious or very serious inaccuracies. Serious inconsistency occurred due to different diagnostic methods and study designs with lower evidence. The inaccuracy was found to be severe or very severe due to wide confidence intervals.

## 4. Discussion

Our meta-analysis found a sixfold higher probability of FP among patients with PC than in the control group. Furthermore, there is a high correlation between the prevalence of FP and the PC in the PC population; however, there was no significant relationship between the FP patient population and PC.

A fatty pancreas is described as a fatty-infiltrated pancreas where adipocytes infiltrate the intralobular and/or interlobular space [14]. The nomenclature of FP is not standardized, but several definitions have been described [53,54].

FP has different possible pathomechanisms. Among these, cellular mechanisms, such as proinflammatory environments, and molecular mechanisms, such as changes in core circadian genes, can be identified [55]. Current knowledge suggests that patients without pancreatic disease can develop FP because of a sedentary lifestyle and obesogenic diet [13]. Fetal or neonatal exposure to a maternal obesogenic diet can predispose to FP in the postnatal period through the imbalance of the endoplasmic reticulum or circadian metabolic patterns [56,57]. In murine studies, excess adiposity increased the levels of triglyceride, cholesterol, total fat, free fatty acids, and proinflammatory cytokines [58], which could lead to an increased risk of FP. Moreover, obesity reduces the anti-inflammatory cytokine interleukin-10 (IL-10), which can contribute to FP [59].

Bi et al. reported in their meta-analysis that FP was associated with an increased risk of metabolic syndrome and its components [12]. Stamm et al. investigated that a 25% increase in pancreatic fat was responsible for a significant increase in the incidence of diabetes and atherosclerosis [60]. A meta-analysis by Garcia et al. [61] showed that pancreatic fat content was increased in patients with diabetes. The association between acute pancreatitis and FP has not been extensively studied but can be a second hit in terms of pancreatic meta-inflammation and chronic damage [62]. Non-alcoholic fatty pancreas disease (NAFPD), a form of FP, is defined as pancreatic fat accumulation associated with obesity and the absence of significant alcohol consumption [50]. Metabolic dysfunction-associated steatotic liver disease (MASLD) is a well-known condition, which was previously named non-alcoholic fatty liver disease (NAFLD). Its propensity for malignant proliferation has already been proven [63]. The pancreas is more susceptible to fat deposition than the liver, resulting in a five times greater risk for fat accumulation [64]. The relationship between MASLD and NAFPD has been described [65,66]. In MASLD, fat accumulates within the hepatocytes, whereas in NAFPD, fat remains in the parenchyma of the pancreas [14].

Our findings suggest that FP is a risk factor for PC. The probability of having FP among patients with PC was higher than in the control group. Additionally, evidence suggests an association between the presence of BMI, chronic pancreatitis, diabetes, pancreas exocrine insufficiency, and FP. All studies included showed a higher number of FP individuals in the case group. Kawamura et al. [47] demonstrated that all patients with PC had FP, and almost a third of the patients in the PC population had an alcohol use disorder, which can contribute to excess adiposity. According to Hoogenboom et al. [40], less than 10% of the cases have a history of alcohol abuse, which may have contributed to the results.

In the proportion analysis, we found FP in more than half of the patients with PC. Kobashi et al. applied strict inclusion criteria that could result in fewer events [48]. In a logistic regression analysis by Lesmana et al. [45] and Khoury et al. [39], FP was the only risk factor that maintained its association with PC (OR 18.03 95%, CI 7.29–44.59 and OR 2.35, 95% CI 1.04–5.33, respectively). No significant difference was found in analysis A, possibly due to the small number of studies included. Based on the studies available, FP may later progress to PC. According to a prospective study by Sepe et al. [36], there was no association between FP and pancreatic adenocarcinoma. The prevalence of PC in patients with FP was 10.6%, whereas the prevalence was 15.0% in patients without FP (*p* = 0.42). Previous data from animal studies and reviews suggest that FP contributes to the development of PC [13,14,52,67].

During our selection process, we chose high-evidence articles and intentionally used cross-sectional and case–control studies, offering advantages in assessing prevalence, associations, and risk factors. Yet, we must recognize the limitations inherent in these designs, including their inability to establish causality and their susceptibility to bias, which should be kept in mind when interpreting and applying our findings.

Data on diagnostic methods were heterogeneous in most studies using EUS. Histological examination is the gold standard for FP diagnosis, but it is not routinely performed due to its invasive nature. The advantages and disadvantages of different methods were summarized by Truong et al. [52]. Due to the difficulty of pancreatic visualization in obese patients, we did not focus on studies that used transabdominal ultrasound as an imaging modality. There are no guidelines for the measurement and diagnosis of FP; however, there is an established score system that is not widely used [68]. Moreover, routine radiological examination does not focus on the pancreatic fat content. The studies included in this meta-analysis did not provide any information on the management of individuals with FP.

Our analysis revealed that the included articles employed a variety of exclusion criteria. Establishing study designs that can effectively determine the connection between fatty pancreas (FP) and pancreatic cancer (PC) is of utmost importance. Such designs necessitate stringent exclusion criteria to account for potential confounding factors, including, but not limited to, diabetes, obesity, family history of PC, chronic pancreatitis, genetic predisposition, alcohol intake, and smoking. In two of the articles, patients with alcohol consumption were excluded [38,45]. However, it is worth noting that only Kobashi et al. excluded other diseases [48] without providing further clarification. Conversely, most of the included studies were retrospective in nature, where the application of strict exclusion criteria was challenging due to limited available information. This limitation may have resulted in an insufficient number of patients for further analysis.

### 4.1. Strengths and Limitations

This is the most comprehensive systematic review and meta-analysis conducted with a rigorous methodology that has analyzed and confirmed the relationship between FP and PC. We followed our pre-registered protocol. In contrast to the previous meta-analysis by Sreedhar et al. in 2019, we applied a more thorough analysis to the topic of PC.

We faced the following limitations: there was considerable statistical heterogeneity due to the clinical heterogeneity of the studies. For instance, different diagnostic methods were used to assess FP. Some studies examined patients who underwent EUS for hepatobiliary indications, whereas other studies selected PC patients who underwent pancreaticoduodenectomy. Cross-sectional and case–control studies are valuable research designs, but they come with several limitations; they cannot establish causality, and selection bias can occur. Additionally, there is no universal definition of FP, which may further complicate the analysis and comparison of the studies. Lastly, there was a moderate and high risk of bias in some of the domains.

### 4.2. Implication for Practice

The utilization of scientific data for community benefits is of paramount importance [69,70]. Our findings suggest that FP should not be ignored. Instead, physicians should be aware of the potential risks of FP and, for early detection of PC, follow the patients accordingly. We suggest determining a standardized definition and examination method for FP as a starting point. Furthermore, specific guidelines on the management of FP and proper scoring systems for high-risk individuals are needed. Additionally, we recommend documenting the presence of FP in the medical records.

### 4.3. Implication for Research

Further prospective longitudinal data collection is necessary to assess the causality relationship between FP and PC in more depth. It is imperative to provide a clear description of whether metabolic syndrome and its components, age, alcohol consumption, smoking, and pancreatitis can influence the development of FP and PC. This would further support the necessity of detecting FP and the importance of follow-up in patients presenting with FP.

## 5. Conclusions

Fatty pancreas is a risk factor for pancreatic cancer.

## Figures and Tables

**Figure 1 cancers-15-04876-f001:**
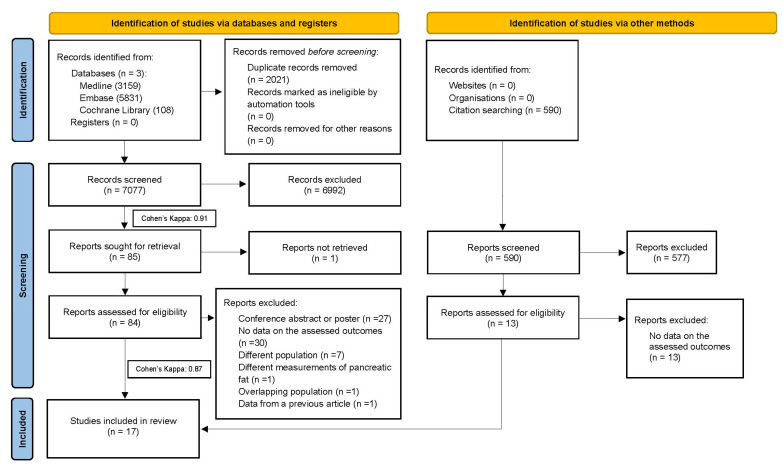
PRISMA 2020 flow diagram of the screening and selection process.

**Figure 2 cancers-15-04876-f002:**
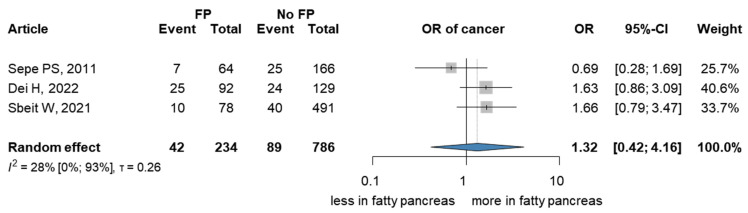
Forest plot representing the presence of PC among patients with FP—Analysis A. FP: fatty pancreas PC: pancreatic cancer, OR: odds ratio, and CI: confidence interval [36,37,38].

**Figure 3 cancers-15-04876-f003:**
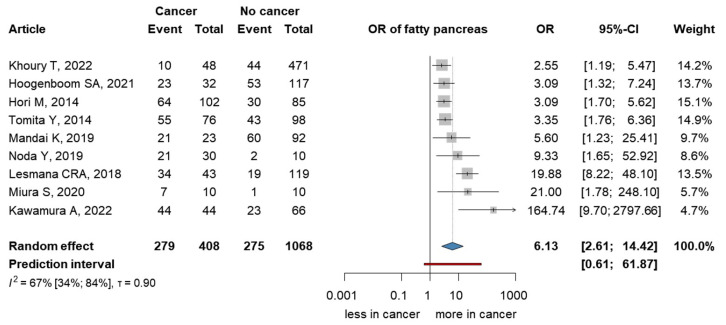
Forest plot representing the presence of FP among patients with PC. FP: fatty pancreas, PC: pancreatic cancer, OR: odds ratio, and CI: confidence interval [39,40,41,42,43,44,45,46,47].

**Figure 4 cancers-15-04876-f004:**
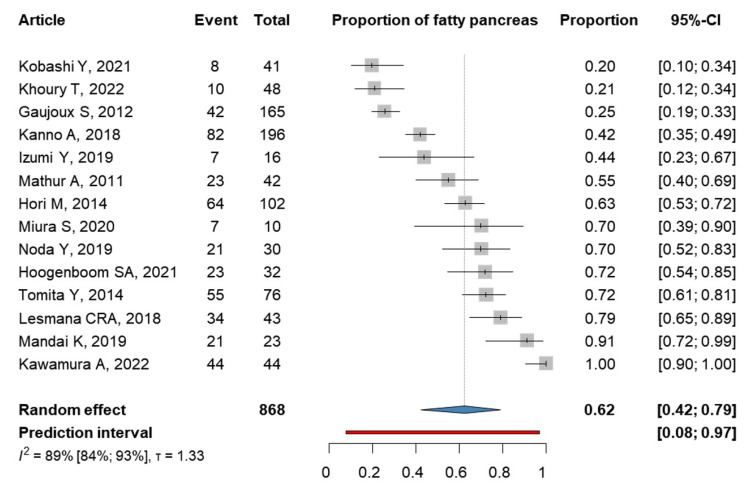
Forest plot on the proportion of FP among PC patients. FP: fatty pancreas, PC: pancreatic cancer, and CI: confidence interval [39,40,41,42,43,44,45,46,47,48,49,50,51,52].

**Table 1 cancers-15-04876-t001:** The main characteristics of the included studies, Analysis A.

Study	Design	Demography				Imaging Modality Used to Assess FP	Total Number of PC Individuals	Total Number of FP in PC Individuals
		Population						
		Country	FP Patients	Control ^c^	Mean Age			
Sepe P. S. et al. (2011) [36]	Cross-sectional	USA	64	198	62.9 ^a^ (13.9) ^b^	EUS	32	7
Sbeit W. et al. (2021) [38]	Cross-sectional	Israel	78	716	62.8 ^d^ (14.1) ^e^	EUS	50	10
Dei H. et al. (2022) [37]	Cross-sectional	Japan	92	172	144 ^f^	CT, Histology	49	25

FP = fatty pancreas; PC = pancreatic cancer; EUS = endoscopic ultrasound; CT = computer tomography; ^a^ = mean; ^b^ = standard deviation; ^c^ = no pancreatic cancer; ^d^ = combined mean age of pancreatic cancer patient group and control group; ^e^ = combined standard deviation of pancreatic cancer patient group and control group; and ^f^ = number of patients that are > 65 years old.

**Table 2 cancers-15-04876-t002:** The main characteristics of the included studies, Analysis B.

Scheme	Design	Demography				Imaging Modality Used to Assess FP	Total Number of PC Individuals with FP	Total Number of FP Individuals in Non-PC Group
		Population						
		Country	PC Patients	Control ^a^	Mean Age			
Mathur A. et al. (2011) [52]	Cross-sectional	USA	42	N/A	46 ^b^ (11) ^c^	CT	23	N/A
Gaujoux S. et al. (2012) [49]	Cross-sectional	USA	165	N/A	71 ^d^ (63–77) ^e^	Histology	42	N/A
Hori M. et al. (2014) [41]	Cross-sectional	Japan	102	85	63.5 ^g^ (56–69) ^i^, 68.0 ^h^ (63–79) ^j^	Histology	64	30
Tomita Y. et al. (2014) [42]	Case–control	Japan	76	98	64.04 ^k^ (11.84) ^l^	Histology	55	43
Kanno A. et al. (2018) [50]	Cross-sectional	Japan	196	N/A	N/A	CT	82	N/A
Lesmana C. et al. (2018) [45]	Cross-sectional	Indonesia	43	119	57 ^b^ (15.9)^c^	EUS	34	19
Izumi Y. et al. (2019) [51]	Cross-sectional	Japan	16	N/A	68.4 ^b^ (52–84 ^f^; 9.1 ^c^)	EUS, Histology	7	N/A
Mandai K. et al. (2019) [43]	Case–control	Japan	23	92	73 ^g^, 73 ^h^ (69–79) ^e^	EUS	21	60
Noda Y. et al. (2019) [44]	Cross-sectional	Japan	30	10	69.9 ^b^ (52–82 ^f^; 7.9 ^c^)	MRI, Histology	21	2
Miura S. et al. (2020) [46]	Cross-sectional	Japan	10	10	68.15 ^k^ (10.23) ^l^	CT	7	1
Hoogenboom S. et al. (2021) [40]	Case–control	USA	32	117	68.49 ^k^ (10.55) ^l^	CT	23	53
Kobashi Y. et al. (2021) [48]	Case–control	Japan	41	N/A	74.8 ^b^ (10.5) ^c^	CT	8	N/A
Kawamura A. et al. (2022) [47]	Cross-sectional	Japan	44	66	64.79 ^k^ (13.52) ^l^	EUS	44	23
Khoury T. et al. (2022) [39]	Cross-sectional	Israel	48	471	63.07 ^k^ (14.01) ^l^	EUS	10	44

FP = fatty pancreas; PC = pancreatic cancer; EUS = endoscopic ultrasound; CT = computer tomography; N/A = not applicable; ^a^ = non-PC patients; ^b^ = mean; ^c^ = standard deviation; ^d^ = median; ^e^ = interquartile range; ^f^ = range; ^g^ = median of case; ^h^ = median of control; ^i^ = interquartile range of case; ^j^ = interquartile range of control; ^k^ = combined mean age of pancreatic cancer patient group and control group; and ^l^ = combined standard deviation of pancreatic cancer patient group and control group.

## Data Availability

The datasets used in this study can be found in the full-text articles that were included in the systematic review and meta-analysis.

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
