# Peer review of "Fatty Pancreas Is a Risk Factor for Pancreatic Cancer: A Systematic Review and Meta-Analysis of 2956 Patients"

_cancers, 2023, doi:10.3390/cancers15194876_

Round 1

Reviewer 1 Report

Dear Editors and Reviewers,

We have reviewed the systematic review and meta-analysis conducted by Mónika Lipp and her colleagues, focusing on the role of fatty pancreas (FP) in the pathogenesis of pancreatic cancer (PC). This study is of significant academic interest as it contributes to our understanding of the factors and mechanisms behind the development of pancreatic cancer.

The researchers conducted a comprehensive review of existing data from MEDLINE, Embase, and CENTRAL, with the primary objective of investigating the association between FP and PC. Their findings indicate that the probability of FP occurrence in PC patients is more than six times higher than in non-PC patients. Therefore, fatty pancreas appears to be a crucial risk factor for the development of pancreatic cancer.

Overall, the paper is well-organized, logically structured, and supported by ample literature evidence. However, there are a few minor issues that the authors should address to enhance the quality of this manuscript.

1. Structural Adjustments: The manuscript's structure should be adjusted to improve its logical flow. Consider reorganizing sections or subsections to enhance the overall coherence of the paper.

2. Abbreviations: The authors should provide full names for some of the abbreviations used within the text to ensure clarity and comprehension for readers.

3. Experimental Design: It is essential to address whether the study's experimental designs, such as cross-sectional and case-control, have any impact on the results. Please discuss the potential implications of these designs on the study's findings and their limitations.

Dear Editors and Reviewers,

We have reviewed the systematic review and meta-analysis conducted by Mónika Lipp and her colleagues, focusing on the role of fatty pancreas (FP) in the pathogenesis of pancreatic cancer (PC). This study is of significant academic interest as it contributes to our understanding of the factors and mechanisms behind the development of pancreatic cancer.

The researchers conducted a comprehensive review of existing data from MEDLINE, Embase, and CENTRAL, with the primary objective of investigating the association between FP and PC. Their findings indicate that the probability of FP occurrence in PC patients is more than six times higher than in non-PC patients. Therefore, fatty pancreas appears to be a crucial risk factor for the development of pancreatic cancer.

Overall, the paper is well-organized, logically structured, and supported by ample literature evidence. However, there are a few minor issues that the authors should address to enhance the quality of this manuscript.

1. Structural Adjustments: The manuscript's structure should be adjusted to improve its logical flow. Consider reorganizing sections or subsections to enhance the overall coherence of the paper.

2. Abbreviations: The authors should provide full names for some of the abbreviations used within the text to ensure clarity and comprehension for readers.

3. Experimental Design: It is essential to address whether the study's experimental designs, such as cross-sectional and case-control, have any impact on the results. Please discuss the potential implications of these designs on the study's findings and their limitations.

Author Response

Response to Reviewer 1 Comments

1. Summary

Thank you for your thoughtful review of our systematic review and meta-analysis “Fatty Pancreas is a Risk Factor for Pancreatic Cancer: A Systematic Review and Meta-Analysis of 2,956 Patients”. We appreciate your valuable feedback and will certainly address the minor issues you've raised to improve the quality of our manuscript. We are committed to making these revisions to enhance the quality of our manuscript. Once again, thank you for taking the time to review our work, we hope receiving a positive evaluation.

2. Point-by-point response to Comments and Suggestions for Authors

1. Structural Adjustments: The manuscript's structure should be adjusted to improve its logical flow. Consider reorganizing sections or subsections to enhance the overall coherence of the paper.

Answer: Thank you for your valuable feedback. We appreciate your constructive suggestions aimed at improving the overall coherence of the paper. Regarding the rearrangement of sections, we share your concern. However, we adhered strictly to the Preferred Reporting Items for Systematic Reviews and Meta-Analyses (PRISMA) guidelines, which limited our flexibility in altering the structure. However, we modified the introduction session.

Action: We modified the introduction part in order to support the understanding. (see page 2, line 69-85)

“At the time of diagnosis, only 10-20% of patients with PC were found in resectable or localized stages [11]. It is crucial to develop a precise screening protocol for the high-risk population unaffected by genetic mutations to facilitate early PC detection.

Due to the high mortality rate as a result of the late diagnosis of PC, non-genetic factors should be determined for early PC detection. FP is a possible option as it can be detected earlier than icterus, which is one of the most common initial features of PC. Fatty pancreas is defined as a significant fatty infiltration of the pancreatic parenchyma and has been associated with increasing age, obesity, metabolic syndrome, non-alcoholic fatty liver disease, and PC [12-15]. The prevalence of FP varies across the studies from 11% to 35% [16,17]. Obesity stands out as a significant risk factor for PC [18]. Furthermore, in-creased pancreatic adiposity contributes to the initiation of PC from FP [14]. Studies by Hori et al. [19] conducted on animals have revealed that obesity and hyperlipidemia exacerbate fatty infiltrated pancreas, consequently accelerating the progression of PC. This process is associated with the increased expression of adipocytokines and genes related to cell proliferation within the pancreatic tissue. Hence, the local production of adipocytokines from adipose tissues in an environment rich in adipose tissue is closely linked to the development of PC [20].”

2. Abbreviations: The authors should provide full names for some of the abbreviations used within the text to ensure clarity and comprehension for readers.

Answer: Thank you for your comment. You are absolutely right.

Action: We have included the complete names of the following acronyms: Preferred Reporting Items for Systematic Reviews and Meta-Analyses (PRISMA), Population, Exposure, Comparator, Outcome(s) (PECO), Condition, Context, Population (CoCoPop), Grading of Recommendations Assessment, Development, and Evaluation (GRADE), Quality of Prognostic Studies (QUIPS) and Joanna Briggs Institute (JBI).

We extended the following sections: (see page 2, line 92-93, 99-101)

“The Preferred Reporting Items for Systematic Reviews and Meta-Analyses (PRISMA) 2020 guideline [18] (see Table S1) and Cochrane Handbook [19] were used.”; ”A Population, Exposure, Comparator, Outcome(s)(PECO) and Condition, Context, Population (CoCoPop) frameworks were used to identify the articles eligible for this study.” (see page 4, line 162-164, 221-224) “Grading of Recommendations Assessment, Development, and Evaluation (GRADE) workgroup recommendation [31] and the GRADE profiler (GRADEpro) tool [32] were applied.” (see page 9, line 221-224) “A [33-35], as well as in those examining patients with PC and the presence of FP [36-44], the Quality of Prognostic Studies (QUIPS) tool was applied. The study attrition domain did not apply to most of the studies. Where the proportion of FP among PC patients was examined, the Joanna Briggs Institute (JBI) tool was applied.”

3. Experimental Design: It is essential to address whether the study's experimental designs, such as cross-sectional and case-control, have any impact on the results. Please discuss the potential implications of these designs on the study's findings and their limitations.

Answer: Thank you for your feedback. Following our systematic search and selection process, it became evident that randomized studies on this topic were lacking due to the nature of the subject matter. As a result, we opted to incorporate articles with the highest available evidence within this field. The decision to utilize cross-sectional and case-control study designs in our research was deliberate, as these designs conferred specific advantages in terms of assessing prevalence, associations, and risk factors. However, it is essential to acknowledge the inherent limitations associated with these designs, including their inability to establish causality and their susceptibility to bias. These limitations should be considered when interpreting and applying the study's findings.

Additionally, we undertook an evaluation of the quality of evidence found in the included articles using the GRADE framework. As a result, we have reached the conclusion that these study designs generally provide lower overall evidence. For a comprehensive breakdown of this evaluation, please refer to Supplementary Materials Table S5-7.

Action: We changed the following sections: (see page 9-10, line 241-242)

“Serious inconsistency occurred due to different diagnostic methods and study designs with lower evidence.” (see page 11 331-333) ”Cross-sectional and case-control studies are valuable research designs, but they come with several limitations; they cannot establish causality and selection bias can occur.”

Reviewer 2 Report

1,A quality assessment of the literature you include is recommended to ensure the accuracy of the conclusions.

2,It is suggested to stratifying or excluding other risk factors that may lead to pancreatic cancer in the population included in the literature.

that's fine

Author Response

Response to Reviewer 2 Comments

1. Summary

We are very grateful for your careful evaluation and constructive criticism of our study. We have revised the manuscript according to the reviewer's comments. Please find our responses to the reviewers’ comments point by point below. Thanks to your suggestions, we think after them, the manuscript has improved a lot. We hope that you will find our corrected and extended paper suitable for publication.

2. Point-by-point response to Comments and Suggestions for Authors

1.     A quality assessment of the literature you include is recommended to ensure the accuracy of the conclusions.

Answer: Thank you for your valuable comment regarding the quality assessment of the literature included in our study. We agree with the importance of this aspect in ensuring the accuracy of our conclusions. After our extensive systematic search, we found eligible studies with moderate to very low evidence based on the certainty of evidence. Please refer to Supplementary Materials Table S5-7.

Action: We edited the “Certainty of evidence”, “Discussion” and “Strengths and limitations” sections: (see page 9-10 line 241-242)

“Serious inconsistency occurred due to different diagnostic methods and study designs with lower evidence used in the studies.” (see page 11 line 296-300) “During our selection process, we chose high-evidence articles and intentionally used cross-sectional and case-control studies, offering advantages in assessing prevalence, associations, and risk factors. Yet, we must recognize the limitations inherent in these designs, including their inability to establish causality and their susceptibility to bias, which should be kept in mind when interpreting and applying our findings.” (see page 11 line 331-333) “Cross-sectional and case-control studies are valuable research designs, but they come with several limitations, they cannot establish causality and selection bias can occur.”

2.     It is suggested to stratifying or excluding other risk factors that may lead to pancreatic cancer in the population included in the literature.

Answer: Thank you. Regarding to the cross-sectional and case-control type of included articles the exclusion of other risk factors were heterogenous across the studies. Based on our opinion a population-based study would be the most ideal of evaluating fatty pancreas and pancreatic cancer. It is also an implication for research.

Action: We modified the “Discussion” section and added a subsection on other risk factors. (see page 11 line 311-321)

“Our analysis revealed that the included articles employed a variety of exclusion criteria. Establishing study designs that can effectively determine the connection between fatty pancreas (FP) and pancreatic cancer (PC) is of utmost importance. Such designs necessitate stringent exclusion criteria to account for potential confounding factors, including but not limited to diabetes, obesity, family history of PC, chronic pancreatitis, genetic predisposition, alcohol intake, and smoking. In two of the articles, patients with alcohol consumption were excluded [38,45]. However, it is worth noting that only Kobashi et al. excluded other diseases [48] without providing further clarification. Conversely, most of the included studies were retrospective in nature, where the application of strict exclusion criteria was challenging due to limited available information. This limitation may have resulted in an insufficient number of patients for further analysis.”

Reviewer 3 Report

Thank you for submitting this comprehensive review study and meta analysis to confirm the relationship between pancreatic cancer and fatty pancreas.  However; the study did not add that much to the literature and to our knowledge regarding this topic. The clinical implications of the study is doubtful given the lacking of novelty in the methods and the serious limitations of the study which the authors has already mentioned some of them in the discussion sections.

The paper was well written and organised.  No major issue.

Author Response

Response to Reviewer 3 Comments

1. Summary

We sincerely appreciate your thorough evaluation and valuable constructive feedback on our study. In response to the comments, we have diligently revised the manuscript. Below, we have provided detailed responses. Thanks to your invaluable suggestions, the manuscript has undergone significant improvement. We are hope that you will find our corrected and expanded paper well-suited for publication.

2. Point-by-point response to Comments and Suggestions for Authors

1. Thank you for submitting this comprehensive review study and meta-analysis to confirm the relationship between pancreatic cancer and fatty pancreas. However, the study did not add that much to the literature and to our knowledge regarding this topic. The clinical implications of the study are doubtful given the lacking of novelty in the methods and the serious limitations of the study which the authors has already mentioned some of them in the discussion sections.

Answer: We greatly appreciate your valuable comment. The association between fatty pancreas (FP) and the development of pancreatic diseases, particularly pancreatic cancer (PC), is a contemporary concern that has been a prominent topic of discussion at international conferences, both in the past and in upcoming events. To date, our understanding of this relationship has heavily relied on review articles. Remarkably, there has been only one systematic review attempting to evaluate the connection between these two entities.

In our perspective, we embarked on a rigorous analysis of FP and PC, employing a meticulous methodology that yielded significant results, previously unreported in this context. The limited depth of exploration into the link between FP and PC is evident, as reflected in the predominantly retrospective nature of the eligible studies, which primarily consisted of case-control and cross-sectional designs. The absence of well-defined guidelines, suitable diagnostic methods, and standardized scoring systems further compounds the challenge of early PC detection, which offers the best chances for improved survival. The implication for both research and practice involves creating guidelines, methods, and scoring systems for everyday use.

Our objective extended beyond mere review; we aimed to construct a meta-analysis that not only synthesizes the current knowledge but also substantiates it with statistical analysis. This approach underscores our commitment to advancing the understanding of the interplay between FP and PC.

Action: We modified the “Discussion” section

(see page 10-12 line 244-350)

“4. Discussion

Our meta-analysis found a sixfold higher probability of FP among patients with PC than in the control group. Furthermore, there is a high correlation between the prevalence of FP with the PC in the PC population; however, there was no significant relationship between the FP patient population and PC.

A fatty pancreas is described as a fatty‐infiltrated pancreas where adipocytes in-filtrate the intralobular and/or interlobular space [14]. The nomenclature of FP is not standardized, but several definitions have been described [53,54].

FP has different possible pathomechanisms. Among these, cellular mechanisms, such as proinflammatory environments, and molecular mechanisms, such as changes in core circadian genes, can be identified [55]. Current knowledge suggests that patients without pancreatic disease can develop FP because of a sedentary lifestyle and obesogenic diet [13]. Fetal or neonatal exposure to a maternal obesogenic diet can pre-dispose to FP in the postnatal period through the imbalance of the endoplasmic reticulum or circadian metabolic patterns [56,57]. In murine studies, excess adiposity increased the levels of triglyceride, cholesterol, total fat, free fatty acids, and proinflammatory cytokines [58], which could lead to an increased risk of FP. Moreover, obesity reduces the anti-inflammatory cytokine interleukin-10 (IL-10), which can contribute to FP [59].

Bi et al. reported in their meta-analysis that FP was associated with an increased risk of metabolic syndrome and its components [12]. Stamm et al. investigated that a 25% increase in pancreatic fat was responsible for a significant increase in the incidence of diabetes and atherosclerosis [60]. A meta-analysis by Garcia et al. [61] showed that pancreatic fat content was increased in patients with diabetes. The association between acute pancreatitis and FP has not been extensively studied but can be a second hit in terms of pancreatic meta-inflammation and chronic damage [62]. Non-alcoholic fatty pancreas disease (NAFPD), a form of FP, is defined as pancreatic fat accumulation associated with obesity and the absence of significant alcohol consumption [50]. Metabolic dysfunction-associated steatotic liver disease (MASLD) is a well-known condition, which previously named as non-alcoholic fatty liver disease (NAFLD). Its propensity for malignant proliferation has already been proven [63]. The pancreas is more susceptible to fat deposition than the liver, resulting in a five times greater risk for fat accumulation [64]. The relationship between MASLD and NAFPD has been described [65,66]. In MASLD, fat accumulates within the hepatocytes, whereas in NAFPD, fat remains in the parenchyma of the pancreas [14].

Our findings suggest that FP is a risk factor for PC. The probability of having FP among patients with PC was higher than in the control group. Additionally, evidence suggests an association between the presence of BMI, chronic pancreatitis, diabetes, pancreas exocrine insufficiency, and FP. All studies included showed a higher number of FP individuals in the case group. Kawamura et al. demonstrated that all patients with PC had FP, and almost a third of the patients in the PC population had an alcohol use disorder, which can contribute to excess adiposity. According to Hoogenboom et al., less than 10% of the cases have a history of alcohol abuse which may have contributed to the results.

In the proportion analysis, we found FP in more than half of the patients with PC. Kobashi et al. applied strict inclusion criteria that could result in fewer events. In a logistic regression analysis by Lesmana et al. [42] and Khoury et al. [36], FP was the only risk factor that maintained its association with PC (OR 18.03 95% CI 7.29–44.59 and OR 2.35, 95% CI 1.04–5.33, respectively). No significant difference was found in analysis A, possibly due to the small number of studies included. Based on the studies available, FP may later progress to PC. According to a prospective study by Sepe et al. [33], there was no association between FP and pancreatic adenocarcinoma. The prevalence of PC in patients with FP was 10.6%, whereas the prevalence was 15.0% in patients without FP (p = 0.42). Previous data from animal studies and reviews suggest that FP contributes to the development of PC [13, 14, 52, 67].

During our selection process, we chose high-evidence articles and intentionally used cross-sectional and case-control studies, offering advantages in assessing prevalence, associations, and risk factors. Yet, we must recognize the limitations inherent in these designs, including their inability to establish causality and their susceptibility to bias, which should be kept in mind when interpreting and applying our findings.

Data on diagnostic methods were heterogenous in most studies using EUS. Histological examination is the gold standard for FP diagnosis, but it is not routinely per-formed due to its invasive nature. The advantages and disadvantages of different methods were summarized by Truong et al. [52]. Due to the difficulty of pancreatic visualization in obese patients, we did not focus on studies that used transabdominal ultrasound as an imaging modality. There are no guidelines for the measurement and diagnosis of FP, however there is an established score system which is not widely used [68]. Moreover, routine radiological examination does not focus on the pancreatic fat content. The studies included in this meta-analysis did not provide any information on the management of individuals with FP.

Our analysis revealed that the included articles employed a variety of exclusion criteria. Establishing study designs that can effectively determine the connection be-tween fatty pancreas (FP) and pancreatic cancer (PC) is of utmost importance. Such designs necessitate stringent exclusion criteria to account for potential confounding factors, including but not limited to diabetes, obesity, family history of PC, chronic pancreatitis, genetic predisposition, alcohol intake, and smoking. In two of the articles, patients with alcohol consumption were excluded [38,45]. However, it is worth noting that only Kobashi et al. excluded other diseases [48] without providing further clarification. Conversely, most of the included studies were retrospective in nature, where the application of strict exclusion criteria was challenging due to limited available information. This limitation may have resulted in an insufficient number of patients for further analysis.

4.1. Strengths and Limitation

This is the most comprehensive systematic review and meta-analysis conducted with a rigorous methodology that has analyzed and confirmed the relationship between FP and PC. We followed our pre-registered protocol. In contrast to the previous me-ta-analysis by Sreedhar et al. in 2019, we applied a more thorough analysis to the topic of PC.

We faced the following limitations: there was considerable statistical heterogeneity due to the clinical heterogeneity of the studies. For instance, different diagnostic methods were used to assess FP. Some studies examined patients who underwent EUS for hepatobiliary indications, whereas other studies selected PC patients who under-went pancreaticoduodenectomy. Cross-sectional and case-control studies are valuable research designs, but they come with several limitations, they cannot establish causality and selection bias can occur. Additionally, there is no universal definition of FP, which may further complicate the analysis and comparison of the studies. Lastly, there was a moderate and high risk of bias in some of the domains.

4.2. Implication for Practice

The utilization of scientific data for community benefits is of paramount im-portance [69,70]. Our findings suggest that FP should not be ignored. Instead, physicians should be aware of the potential risks of FP and for early detection of PC follow the patients accordingly. We suggest determining a standardized definition and ex-amination method for FP as a starting point. Furthermore, specific guidelines on the management of FP and proper scoring systems for high-risk individuals are needed. Additionally, we recommend documenting the presence of FP in the medical records.

4.3. Implication for Research

Further prospective longitudinal data collection is necessary to assess the causality relationship between FP and PC in more depth. It is imperative to provide a clear de-scription of whether metabolic syndrome and its components, age, alcohol consumption, smoking, and pancreatitis can influence the development of FP and PC. This would further support the necessity of detecting FP and the importance of follow-up in patients presenting with FP.”

Round 2

Reviewer 2 Report

It will be convinced if you can find more dates where FP patients are likely to have PC, because it's prospective. 

Reviewer 3 Report

The revised manuscript was substantially improved and I found it is acceptable for publication.